# The Regular Consumption of Nuts Is Associated with a Lower Prevalence of Abdominal Obesity and Metabolic Syndrome in Older People from the North of Spain

**DOI:** 10.3390/ijerph19031256

**Published:** 2022-01-23

**Authors:** Gloria Cubas-Basterrechea, Iñaki Elío, Sandra Sumalla-Cano, Silvia Aparicio-Obregón, Carolina Teresa González-Antón, Pedro Muñoz-Cacho

**Affiliations:** 1Dietetic Section, Hospital Universitario “Marqués de Valdecilla”, 39008 Santander, Spain; 2Research Group on Foods, Nutritional Biochemistry and Health, Universidad Europea del Atlántico, 39011 Santander, Spain; inaki.elio@uneatlantico.es (I.E.); sandra.sumalla@uneatlantico.es (S.S.-C.); silvia.aparicio@uneatlantico.es (S.A.-O.); 3Department of Health, Nutrition and Sport, Iberoamerican Internarional University, Campeche 24560, Mexico; 4NEXO Multidisciplinar Center, Department of Nutrition and Dietetic, 04004 Almeria, Spain; carolinaglan@gmail.com; 5Teaching Department of Primary Care Management, Cantabrian Health Service, IDIVAL, 39011 Santander, Spain; pedro.munoz@scsalud.es

**Keywords:** elderly, nuts, metabolic syndrome X, aging, Spain

## Abstract

Background: The aim of this study was to relate the adherence to nut consumption (30 g) three or more days per week to the prevalence of abdominal obesity and metabolic syndrome (MetS) in an elderly population from the north of Spain. Methods: The study consists of an observational, descriptive, cross-sectional, and correlational study conducted in 556 non-institutionalised individuals between 65 and 79 years of age. To define the consumption recommendation of nuts the indication of the questionnaire MEDAS-14 was followed. The diagnosis of MetS was conducted using the International Diabetes Federation (IDF) criteria. Results: In 264 subjects aged 71.9 (SD: ±4.2) years old, 39% of whom were men, the adherence to nut consumption recommendations was 40.2%. Of these individuals, 79.5% had abdominal obesity. The prevalence of MetS was 40.2%, being 47.6% in men and 35.4% in women (*p* < 0.05). A nut consumption lower than recommended was associated with a 19% higher prevalence of abdominal obesity (Prevalence Ratio: 1.19; 95% CI: 1.03−1.37; *p* < 0.05) and a 61% higher prevalence of MetS (Prevalence Ratio: 1.61; 95% CI: 1.16−2.25; *p* = 0.005) compared to a consumption of ≥3 servings per week. Conclusion: An inverse relationship was established between nut consumption and the prevalence of abdominal obesity and metabolic syndrome.

## 1. Introduction

The ageing of the global population has acquired significant dimensions, meaning that any issue related to the health of this population has great repercussions. In Spain, according to data from the National Statistics Institute (INE) [1], elderly people (65 years and over) form 19.8% of the total population. Within Spain, the community of Cantabria is the fifth oldest community in Spain, with a percentage of older people of 22.7% [1]. In Santander, the capital of Cantabria and the place where the study was carried out, the percentage is even higher, reaching 24.2% in 2021 [2]. These numbers are sufficiently relevant to propose research studies to improve the public health of this population group.

Achieving greater longevity should be understood not only as the maximum possible length of human life, but it also indicates a higher quality of life in terms of good health [3]. “Healthy ageing”, as proposed by the WHO in 2015 [4] is defined as “the process of developing and maintaining the functional capacity that enables well-being in old age”. With ageing comes a generalized and progressive deterioration of biological functions, a greater vulnerability to the environment and an increased risk of disease and death, with a different rate for each individual [5,6]. This is because genetics contributes to 25% of the changes that occur in ageing (physiological) and the remaining 75% (epigenetic ageing) of changes due to accumulated diseases and environmental and lifestyle factors, such as diet, which are modifiable [7]. In this regard, there is strong evidence that healthy dietary patterns can modulate atherogenesis directly or through an effect on classical risk factors such as plasma lipids, blood pressure and blood glucose concentration, all of which are associated with MetS [8]. MetS is a set of risk factors associated with abdominal obesity and insulin resistance, characterised by high blood pressure, hyperglycaemia and lipid alterations such as hypertriglyceridemia and lower HDL-c, which will facilitate the development of type 2 diabetes mellitus (DM2) and cardiovascular diseases (CVD) [9]. Nuts are rich in macronutrients and micronutrients that will promote health because they contain a series of bioactive compounds. These include fat-soluble compounds such as unsaturated fats (monounsaturated fatty acids: MUFAs, polyunsaturated fatty acids: PUFAs), tocols (tocopherols and tocotrienols), phytosterols, sphingolipids, carotenoids and chlorophylls, vegetable protein, dietary fibre, water-soluble vitamins (group B), minerals (magnesium, potassium and calcium), and phenolic compounds, which are divided into flavonoids, phenolic acids, stilbenes (resveratrol), hydrolysable tannins (derived from ellagic acid and gallic acid) and condensed tannins, which will be determinant in a series of beneficial biological effects [10,11,12]. This nutritional composition of nuts makes them of particular interest to older people by improving the quality of their diet, nutritional status [13] and levels of oxidative stress and inflammation [14]. Therefore, the consumption of a healthy diet as part of the Mediterranean diet (rich in vegetables, fruits such as strawberries, virgin olive oil, legumes, white meats, fish and whole grains) in which nuts are regularly included, as a food rich in unsaturated fats replacing foods rich in saturated fats or industrially produced trans fats, is a modifiable factor that can facilitate healthy ageing by reducing the risk of developing pathologies with a high prevalence in this population group, and that are related to cardiovascular diseases such as MetS [15,16,17] and even associated with psychiatric diseases such as dementia thanks to an improvement in cognitive function [18,19]. 

The aim of this study was to relate the adherence to nut consumption (30 g) three or more days per week, with the prevalence of abdominal obesity and metabolic syndrome (MetS) in an elderly population from the north of Spain.

## 2. Materials and Methods

### 2.1. Study Design

This is an observational, descriptive, cross-sectional and correlational study to assess adherence to the recommended consumption of nuts in the elderly population, to determine the prevalence of abdominal obesity and MetS and to analyse whether there is any relationship between them, and also to determine nutritional status through the Body Mass Index (BMI). The scientific-technical validation of the study was obtained from the Primary Care Management of the Cantabrian Health Service (CHS).

### 2.2. Participants

The study population was non-institutionalised elderly people with ages between 65 and 79 years, belonging to the quota of 4 doctors of 3 Primary Care Centres (PCC) in Santander (Cantabria) of the CHS. According to data published by the Cantabrian Institute of Statistics (ICANE) on 28 January 2021, the population with these characteristics amounted to 29,604 individuals [2]. For the calculation of the sample number, a difference of 20% in the consumption of ≥3 servings of nuts per week was estimated between patients with METs and those without METs and the same between those with abdominal obesity and those without, assuming an alpha risk of 5% and a beta risk of 20% in a bilateral contrast. This assumes at least 96 subjects per group, that is, 192 in total. The Granmo v.7.12 program for finite populations [20] was used to calculate the sample. 

To obtain this sample, a three-layer sampling was conducted: firstly, the three PCCs in Santander with the highest number of patients over 65 years of age were chosen; subsequently, a medical quota was chosen from each of the three centres on a purposive basis; that is, it was offered to the coordinator of each of these PCCs and a second medical quota proposed by the PCC coordinator was chosen from the centre that had the highest number of patients in this age group; finally, a random sample of patients stratified by sex and age (65−79 years), chosen by systematic sampling, was selected.

The study started with 556 individuals, of which 239 were ineligible; they were withdrawn by doctors as they met the following exclusion criteria: neurological; psychiatric/psychological pathologies; physical stability problems; mental and cognitive impairment (Pfeiffer test > 4 errors) [21] and weight change (> or <10%) in the last 12 months; because they could not be reached by telephone; or because they were unable or unwilling to participate in the study. Of the remaining 317 individuals, there were 53 individuals who were not selected due to missing biochemical blood analysis and/or medication data for the last 12 months for MetS diagnostic parameters according to the International Diabetes Federation (IDF) criteria [22]. Therefore, after random and systematic sampling and after applying the selection and substitution criteria, a final sample of 264 participants (male: 39% and female: 61%) was obtained (Figure 1).

### 2.3. Sociodemographic Variables

Sociodemographic characteristics were analysed, including sex, age, marital status, type of cohabitation and level of education. The following three age groups were established: 65−69 years, 70−74 years and 75−79 years. Marital status included the following four levels: married/partnered, separated, widowed and single. Type of cohabitation was divided into couple, with relatives, with a carer, alone or in a shared flat. Educational level was classified as university, secondary school, primary school and incomplete primary school. 

### 2.4. Body Mass Index Levels

The calculation of the BMI was the result of the quotient between weight and height squared. According to the SEEDO 2000 classification [23], the following 3 levels of nutritional status were established according to their corresponding BMI thresholds: normal weight (18.5−24.9), overweight (25−29.9) and obesity (30−49.9). 

### 2.5. Diagnosis of Metabolic Syndrome according to IDF Criteria 

To make the diagnosis of MetS, the IDF criteria [22] were used, according to which abdominal obesity is an essential diagnostic parameter (waist circumference for European individuals: ≥94 cm in men and ≥80 cm in women) and also 2 or more of the following parameters must also be present: arterial hypertension (≥130/85 mmHg, being on treatment or diagnosed), fasting hyperglycaemia (≥100 mg/dL or previous diagnosis of DM2 or treatment), hypertriglyceridemia (≥150 mg/dL or on treatment) and low HDL-c (<40 mg/dL in men and <50 mg/dL in women or on treatment).

### 2.6. Instruments

#### 2.6.1. Adherence to Recommendations for Nut Consumption

To define the recommendation for nut consumption in older people, the item of the MEDAS-14 questionnaire related to nuts was considered, which recommends consumption of one serving (30 g) 3 or more times a week [24]. Study participants were asked the following question: How many times a week do you consume one serving of nuts (30 g)? The nuts considered in the study included almonds, Brazil nuts, cashews, hazelnuts, pine nuts, pistachios and walnuts. 

#### 2.6.2. Body Mass Index Assessment

For the determination of the BMI, weight was measured with the SECA^®^ 711 scale model (SECA, Hamburg, Germany) Height was measured directly with the SECA^®^ 220 stadiometer (SECA, Hamburg, Germany) and the SECA^®^ 213 portable model (SECA, Hamburg, Germany), and indirectly by measuring the knee−heel distance, measured with a stadiometer, using the formula of Chumlea et al. that relates age to knee height [25]. 

#### 2.6.3. Assessment of Diagnostic Parameters for Metabolic Syndrome 

To measure waist circumference and subsequently assess the presence of abdominal obesity, a SECA model 203 ergonomic tape (SECA, Hamburg, Germany) with millimetric precision was used. For the diagnosis of hypertension, blood pressure was measured with the OMRON M3 comfort^®^ automatic arm blood pressure monitor (Omron, Shimogyo-ku, Kyoto, Japan). The addition of one or more medications for the treatment of hypertension on the health card was also useful. The diagnosis of fasting hyperglycaemia was made based on fasting blood glucose values and prescribed medication for DM2 taken from the health card. Hypertriglyceridemia could be diagnosed through blood triglyceride results and triglyceride medications included in the health card. Low HDL-c was diagnosed using the HDL-c blood data from the health card.

### 2.7. Procedure

To each individual selected, a postal letter was sent from the CHS informing them about the study and a few days later they were called by telephone to request their participation. Once they had accepted, they were summoned to the corresponding PCC and two informed consent forms were handed out (one to be signed by the participant to give consent and the other signed by the researcher, for possible revocation of consent). Afterward, the appropriate procedures were then carried out to obtain the data on adherence to the recommended nut consumption, the prevalence of abdominal obesity and MetS, and the assessment of BMI, as indicated below.

#### 2.7.1. Nut Consumption

The dietician-nutritionists, previously trained, asked the participants in the study the question above mentioned to know about their consumption of nuts, and to facilitate their understanding, they were helped by various portions of nuts or photographs.

#### 2.7.2. Assessment of Nutritional Status Using BMI

Anthropometric measurements were taken in the morning, with the individuals barefoot and wearing basic clothing. Two measurements were taken and the mean was calculated. For the weight measurement, the participant was in an erect and relaxed position facing the scale and their eyes were fixed on a horizontal plane. For the height measurement, the height at maximum extension technique was used, with the head in a horizontal position in the Frankfort plane. In the case of spinal pathologies or difficulties in maintaining balance, the distance between the sole of the foot and the upper limit of the patella was measured and the formula created by Chumlea et al. [25] was applied to determine the height in this type of subjects.

#### 2.7.3. Assessment of Diagnostic Parameters for Metabolic Syndrome 

For the determination of abdominal obesity, the waist circumference (midline between the lower costal margin and the upper edge of the iliac crest, in standing position) was measured [26]. The anatomical reference point was considered to be 2.5 cm above the umbilicus, as this has been shown in the elderly population to be the best associated with abdominal adipose tissue measured using dual-energy X-ray absorptiometry (DXA) and therefore the best indicator of adipose tissue [27]. Two measurements were performed, and the mean was calculated. If the difference between the two measurements was >2%, another measurement was performed and the median was taken as valid data [28]. 

Blood pressure was measured after 2 to 3 min of rest, in a seated position, on the dominant arm. The mean of three measurements was calculated and at least 1 min was allowed to elapse between the two measurements. The rest of the MetS diagnostic parameters were obtained by accessing the biochemical blood analysis and medication data from each individual’s CHS health card. Finally, a diagnosis of MetS was made when the waist circumference was greater than the values indicated according to the IDF criteria and at least two of the diagnostic parameters for MetS were also met.

### 2.8. Statistical Analysis

The SPSS 25 software was used to analyse the data (IBM Corp. Released 2017. IBM SPSS Statistics for Windows, Version 25.0 Armonk, NY, USA: IBM Corp.) and Epidat 4.2, July 2016 (Consellería de Sanidade, Xunta de Galicia, Spain).

Qualitative variables were described by calculating frequencies and percentages. To establish the association between an independent variable (consumption of nuts ≥ 3 servings/week, abdominal obesity, prevalence of MetS) and a dichotomous variable (gender) or more than one category (age groups), the Pearson’s chi-square statistical test was performed. If the qualitative variable has several categories that do not have an order (types of nutritional status according to BMI) to compare the results by gender or age groups, the chi-square test of goodness of fit was used, with the Epidat 4.2 program (Consellería de Sanidade, Xunta de Galicia, Spain). If the qualitative variable with several categories has an order (number of variables of diagnosis of MetS), the chi-square test of trend allowed us to calculate if there was any statistically significant difference by gender or age groups in each category.

In the case of quantitative variables, the Kolmogorov−Smirnov (K−S) non-parametric test of normality was used to determine whether the distribution was normal. The variable with a normal distribution (waist circumference) was described with the mean and standard deviation (SD). To compare two categories (gender) and two independent groups, Student’s *t*-test was performed. To compare with more than two categories (age groups), the analysis of variance test (ANOVA) was performed.

Variables with a non-normal distribution (BMI, number of MetS diagnostic variables) were described with median and interquartile range (IQR). To compare two categories (gender) and two independent groups, the non-parametric Mann−Whitney test was performed. To compare with a variable of more than two categories (age groups) the non-parametric Kruskal−Wallis test was used.

Finally, comparisons between variables were made to look for the possible association between the consumption of nuts adapted to the recommendations and the prevalence of abdominal obesity and MetS. Prevalence ratio (PR) was used as a measure of association and calculated using log-binomial regression.

## 3. Results

### 3.1. Adherence to Recommended Nut Consumption

Of the individuals (N = 264), 40.2% consumed nuts according to the recommendations (one serving between 3 and 7 days a week) and 59.8% consumed a lower amount. There were no statistically significant differences by sex and age groups, marital status, type of cohabitation and level of education (Table 1).

On the other hand, a significant relationship between nut consumption and BMI ranges was observed (*p* = 0.035), with an increase in the prevalence of the consumption of <3 servings of nuts per week as the BMI increases: normal weight (48.4%), overweight (59.8%) and obese (70.6%) (*p* for trend: *p* = 0.010). Regarding the relationship between nut consumption and health-related variables, a significant difference (*p* = 0.002) was only observed in the case of individuals diagnosed with hypertriglyceridemia, the prevalence being higher in individuals who consumed <3 servings of nuts per week (78.2%) compared to those who consumed ≥3 servings per week (21.8%) (Table 1).

### 3.2. Prevalence of Abdominal Obesity

The mean waist circumference in men (N = 103) was 102.6 (±DE:10.9) and in women (N = 161), it was 90.7(±DE:13.6) (*p* < 0.001). The mean waist circumference values were much higher than the diagnostic values for abdominal obesity (≥94 cm in men and ≥80 cm in women).

The prevalence of abdominal obesity in the current study was 78.8%; being lower in men (77.7%) compared to women (79.5%), with no significant difference (Appendix A). There were also no significant differences by age group.

### 3.3. Prevalence of Nutritional Status according to BMI

In the sample, 24.2% were normal weight, 50.0% overweight and 25.8% obese (*p* = 0.001). By gender, men had a lower prevalence of normal weight (12.6%) than women (31.7%) (*p* < 0.001). In contrast, men had a higher prevalence of overweight (60.2%) compared to women (43.5%) (*p* < 0.01) and obesity (27.2%) compared to women (24.8%), with no significant difference (Appendix A). No significant differences were established by age group.

### 3.4. Prevalence of Metabolic Syndrome

As shown in Appendix A, the prevalence of MetS was 40.2%. By gender, the prevalence in men (N = 103) was 47.6% and in women (N = 161) 35.4%, with a statistically significant difference (*p* < 0.05). The prevalence of MetS varied little between the different age groups, with no statistically significant differences.

The highest prevalence in both sexes was hypertension (79.5%): 86.4% in men and 75.2% in women (*p* < 0.05). The prevalence of hyperglycaemia (31.8%) was also higher in men (45.6%) compared to women (23.0%) (*p* < 0.001). The prevalence of hypertriglyceridemia was much lower (23.4%) and similar in men (20.4%) and women (23.6%). Finally, the prevalence of low HDL-c was 19.7% (20.4% in men and 19.3% in women) (Appendix A).

The majority of individuals (38.6%) had one MetS diagnostic variable other than abdominal obesity. Next, 27.6% had two diagnostic variables; the percentages of individuals with no or three MetS diagnostic variables were similar (14.4 and 15.2%, respectively). Lastly, the percentage of individuals with four diagnostic parameters was the lowest (4.2%). There was a significant difference in the percentage of the number of MetS diagnostic variables (hypertension, hyperglycaemia, hypertriglyceridemia, low HDL-c) by gender (*p* < 0.001). Females had a higher percentage of zero or one variable (19.9 and 46.0%, respectively) compared to males (5.8 and 27.2%, respectively) (*p* < 0.005); in contrast, males had a higher percentage of two, three and four MetS diagnostic variables (35.9, 22.3 and 8.7%, respectively) compared to females (22.4, 10.6 and 1.2%, respectively) respectively (Appendix A).

As shown in Table 2, the prevalence of MetS increases with BMI (*p* < 0.001). By gender, the prevalence of MetS in men was higher at all the BMI levels compared to women, with the difference being significant in normal weight and obesity.

### 3.5. Association between the Adherence to Recommended Nut Consumption and Prevalence of Abdominal Obesity

The individuals who consumed one serving (30 g) of nuts three or more times a week had a prevalence of abdominal obesity of 70.8%, compared to the consumers of nuts less than three times a week which had a 13.4% higher prevalence of abdominal obesity (84.2%) (*p* < 0.01). 

In addition, it was found that the proportion of people with abdominal obesity was 19% higher among those who consumed a portion of nuts less than three times a week compared to those who consumed nuts three or more times a week (PR: 1.19; 95% CI: 1.03–1.37; *p* < 0.05) (Table 3).

### 3.6. Association between Adherence to the Recommended Consumption of Nuts and Prevalence of Metabolic Syndrome

The individuals who consumed one serving (30 g) of nuts three or more times a week had a prevalence of MetS of 30.2%, compared to the consumers of nuts less than three times a week which had a 16.6% higher prevalence of METs (46.8%) (*p* < 0.01). 

It was also found that the proportion of people with MetS was 61% higher among those who consumed a portion of nuts less than three times a week compared to those who consumed nuts three or more times a week (PR: 1.61; 95% CI: 1.16–2.25; *p* = 0.005) (Table 3).

## 4. Discussion

### 4.1. Adherence to Recommended Nut Consumption

Nuts stand out for being the second food, after olive oil, that provides unsaturated fatty acids (MUFAs and PUFAs) with anti-inflammatory and vaso-protective activity [29]. Omega-3 fatty acids are a family of essential and biologically relevant fatty acids that belong to the PUFAs, being mostly available in nuts, alfa-linolenic acid (ALA), which is metabolised to eicosapentaenoic acid and docosahexaenoic acid, with anti-inflammatory properties. In this regard, a recent meta-analysis published in 2021 by Naghshi et al. [30] found that a high intake of ALA was associated with a reduced risk of mortality from all causes, CVD and coronary heart diseases. Additionally, the nuts contain phytosterols, which have an inverse relationship with blood cholesterol levels [31] and other bioactive compounds with antioxidant properties such as polyphenols and vitamin E, all of which play an important role in the prevention of cardiovascular disease [32]. Moreover, ageing is associated with oxidative damage and a pro-inflammatory state; therefore, age can be considered a risk factor for CVD [33,34]. In this regard, it has been shown that regular nut consumption is associated with reductions in some, but not all, markers of oxidative stress and inflammation [14]. However, only 40.2% of the elderly participants in the study consumed three or more servings (30 g) of nuts per week; there were no significant differences in consumption according to the different socio-demographic variables studied (sex, age groups, marital status, type of cohabitation and level of education) (Table 1). However, a significant association was observed by increasing the prevalence of the consumption of <3 servings per week of nuts with an increasing BMI (*p* of trend: 0.010) (Table 1), which seems to indicate that, especially in overweight and obese elderly people, the consumption of nuts was mainly less frequent than recommended, and thus decreasing the adherence to the Mediterranean diet. A possible reason for this low nut consumption is the belief in the older population that the composition of nuts, rich in fatty acids, is associated with weight gain. However, a systematic review and meta-analysis of prospective cohorts and randomized controlled trials published in 2021 [35] evaluating the impact of nut consumption on adiposity measures concluded that, based on current evidence, health professionals and dietary guidelines can advise a daily consumption of a 42.5 g serving, except for people with nut allergies, without an increase in adiposity or concern for an adverse on body weight control. According to previous studies carried out in other populations in the Mediterranean basin, it seems that a lower than recommended consumption of nuts is common in older people, since in a recent study [36], in individuals aged 60−69 years, the percentage of adherence to the recommendations was 34% and in individuals aged 70−79 years it was 32%, and on the other hand, in a population aged over 65 years living in Sicily, the percentage of individuals with a low consumption of nuts (average 4.3 g/day) was significantly higher (21.9%) than those with a high nut consumption (mean 39.7 g/day), which stood at 16.4% (*p* = 0.004) [37]. The low consumption of nuts obtained in the study coincides with the results of the Spanish Food Consumption Report [38], as Cantabria is the second community with the lowest consumption of this type of food, with a consumption below the average (3.35 kg/person/year). 

Another possible reason for the low consumption of nuts in elderly people may be related to chewing problems and dysphagia that are very common in this population group, however, should facilitate the daily consumption of small amounts (between 28.3 to 42.5 g) [35], through techniques such as crushing and incorporating them into foods that are easy to chew and swallow such as yogurts. In this way, the population mentioned before will take advantage of the beneficial properties demonstrated in the PREDIMED study [39], according to which the consumption of a Mediterranean diet supplemented with 45 g of nuts daily in an elderly population at high cardiovascular risk reduced the risk of myocardial infarction and cerebrovascular infarction and mortality from these pathologies by up to 30%, associated with improvements in the components of MetS.

### 4.2. Prevalence of the Metabolic Syndrome

The prevalence of MetS, according to the IDF criteria [22], was 40.2%. This prevalence resembles that obtained for the Spanish population over 65 years of age in the ENRICA study with 42.3% [40] and is quite distant from that obtained in other parts of the world: Mexico (72.9%) [41], United States (54.9 ± 1.7%) [42], Iran (51.7%) [43] and Russia (34.7%) [44].

In the HERMEX study [45], the prevalence in individuals aged 65–74 years was 59.6% and in those aged 75−79 years it was 64.8%, increasing with age. It is common for the prevalence of MetS to be higher with increasing age because, as it is associated with the ageing process, the prevalence of the number of components of MetS tends to increase. However, in the current study, no significant differences were observed in the prevalence of MetS by age group; but differences by gender were observed.

The prevalence of MetS obtained was higher in men than in women (47.6% vs. 35.4%) (*p* < 0.05). These results contrast with a large majority of studies, obtained using different methods of diagnosing MetS, carried out in various countries worldwide in the elderly population, in which the prevalence obtained was higher in women: in Korea [46]: 50% vs. 36.4%; in China [47]: 47.55% vs. 39.74%; in India [48]: 50.9% vs. 34.4%; in Ecuador [49]: 66.9% vs. 47.9%; in Brazil [50]: 65.6% vs. 60.3%; in Russia [44]: 41.7% vs. 26.8%; in Portugal [51]: 40% vs. 22%; in Finland [52]: 47.8% vs. 37.2%; in Mediterranean islands (Malta, Cyprus and Greece) [53]: 35% vs. 24%; and in Spain in the DARIOS study [54]: 52.5% vs. 42.2%, the ENRICA study [40]: 44.5% vs. 39.5% and in the HERMEX study [45]. However, in other studies the prevalence in older people was higher in men, such as in the Mexican population where the prevalence was 75.7% in men and 70.4% in women [41], in France with ages between 55−74 years (40.3% vs. 34.4%) with no significant difference [55] and in the Dutch population taking into account sex, age and BMI together [56]. 

The higher prevalence of MetS in men obtained in the study does not seem to be related to the accumulation of android fat, with a higher prevalence of abdominal obesity, as there were no statistically significant differences by sex in this parameter, although in both sexes it was high and higher in women (79. 5% vs. 77.7%), coinciding with the results of the ENRICA study [40], in which the prevalence of abdominal obesity is higher in women (94.2%) than in men (82.4%), due to the fact that central adiposity is very common in women after menopause [57]. The high prevalence of abdominal obesity in older women is due to the fact that at this stage of life the decrease in estrogen levels leads to the hypertrophy of adipocytes and a rapid increase in visceral adipose tissue, which is associated with insulin resistance and an inflammatory and prothrombotic state [58].

The higher prevalence of MetS in men could be related to differences in BMI (Appendix A); as men had a higher prevalence of overweight (60.2%) compared to women (43.5%) (*p* < 0.01), a similar but higher prevalence of obesity in men (27.2% vs. 24.8%) and a lower prevalence of normal weight in men (12.6% vs. 31.7%) (*p* < 0.001). All of this linked with a high prevalence of abdominal obesity in men (77.7%), will facilitate adipose tissue dysfunction and insulin resistance and the development of metabolic comorbidities and, therefore, a higher prevalence of MetS [59] and CVD.

However, it was found that although the prevalence of MetS increases with BMI for both sexes, the prevalence of MetS in normal weight and obese men was significantly higher than in women (Table 2). In this way, it was shown that MetS was 84% more frequent in men than in women (PR: 1.84; 95% CI: 1.39–2.45; *p* < 0.001). 

Therefore, it seems that not only factors related to BMI are involved in the higher prevalence of METs in men, associated with men with a higher percentage of two, three and four diagnostic variables compared to women (Appendix A), other known factors involved in the development of METs should also be considered. It is recognized that the increasing prevalence of MetS in older people is influenced by age, as well as by gender, by an altered testosterone/estrogen balance [56], and also by various factors such as the consumption of an unhealthy diet; overweight-obesity; a sedentary lifestyle (smoking, stress, alcoholism); socio-economic aspects (culture-education, economic level) [60]; social relationships, since if a friend, sibling or partner is obese, the likelihood of an individual being obese increases [61]; place of residence, whether rural or urban [51]; and even pathologies such as depression, often associated with ageing, can be affected by being associated with obesity, an unhealthy diet and physical inactivity [62]. All of these factors may affect individuals and men and women differently and may determine gender differences in the prevalence of MetS.

### 4.3. Relationship between Adherence to Recommended Nut Consumption and Prevalence of Abdominal Obesity and Metabolic Syndrome

The study found that the consumption of a portion of nuts < 3 times per week was associated with a 19% higher prevalence of abdominal obesity and a 61% higher prevalence of MetS compared to consumption ≥ 3 times a week.

Since this is an observational study, no causal relationship can be extracted. However, there are recent longitudinal studies that confirm the benefits of regular nut consumption. Thus, in the PREDIMED study, conducted in older people at cardiovascular risk, it was shown that the consumption of more than three servings of nuts per week was associated with a 32 and 26% lower prevalence of abdominal obesity and MetS, respectively, compared to those who did not consume nuts [63]. Similar results were obtained in the University of Navarra follow-up study (SUN) [64] in which women who consumed more than two servings of nuts per week had a lower risk of MetS compared to those who consumed less. Even in countries outside the Mediterranean basin, there has been interest in the influence of nut consumption, particularly walnuts, on the development of MetS. According to Hosseinpour-Niazi et al. [65], the consumption of ≥5 servings/week of walnuts reduced the incidence of MetS compared to when consumption was ≤1 serving/week (OR: 0.68, 95% CI: 0.44–0.91, *p* for trend:0.03). Furthermore, for each additional serving consumed per week, the incidence of MetS decreased by 3%. Additionally, in another study by Hassannejad et al. [66], it was concluded that the regular consumption of nuts was negatively associated with the severity and risk of developing MetS. 

These results are associated with the beneficial effects of nut composition on various parameters related to metabolic syndrome. In this regard, daily consumption of nuts over 5 years of follow-up has been observed to reduce waist circumference gain and decrease body weight [67], related to their satiating effect, incomplete digestion in the gut and lower amount of metabolizable energy. In addition, nuts appear to have a preventive effect on the development of overweight/obesity, despite what might be assumed from their fatty acid-rich nutritional composition, as individuals who consume nuts frequently tend to eat less red meat and processed meat, facilitating adherence to a healthy diet [68]. A meta-analysis of prospective studies [69] showed a 15% reduction in the risk of high blood pressure when comparing extreme nut consumption (0−37 g/day), which is possibly associated with a reduction in inflammation and an improvement in the endothelial dysfunction attributable to their composition (rich in L-arginine, MUFAs, PUFAs, polyphenols and non-sodium minerals). Additionally, a recent study [37] showed an inverse association between high nut consumption (mean 39.7 g/day) and lower odds of high blood pressure (OR = 0.61, 95% CI: 0.46–0.80). Furthermore, the reduction in hypertension through regular nut intake has a biological explanation, as it implies a lower risk of body adiposity, as indicated above and the fact that 75% of the incidence of hypertension is related to obesity [70]. In a meta-analysis of 61 randomized controlled trials [71], it was found that a daily consumption of nuts shows a reduction in the blood levels of total cholesterol, LDL-c and triglycerides, but has no effect on HDL-c levels, which are more associated with the frequency and number of servings consumed rather than the type of nut. In the present study, it was also found that the prevalence of hypertriglyceridemia was lower in individuals who consumed ≥3 servings of nuts per week (21.8%) compared to those who consumed <3 servings per week (78.2%) (*p* = 0.002). Additionally, the composition of nuts that are rich in PUFAs, a dietary fibre that delays gastric emptying, reduces intestinal absorption and increases satiety; and components such as ellagic acid with anti-inflammatory and anti-oxidative properties, explains the greater glycaemic control and lower incidence of DM2 [72]. Nuts are one of the most important sources of omega-3 fatty acids. In this regard, a meta-analysis of 45 clinical trials found that supplementation with omega-3 fatty acids was associated with improved lipid profile, inflammation, and glycaemia in people with diabetes [73].

The results obtained, being an observational study with a small number of individuals, have their limitations, particularly in terms of being able to draw cause-effect conclusions. It is necessary to add as a limitation in this study that the association of nut consumption with abdominal obesity and METs has not been controlled for possible confounding variables. Finally, another limitation is that no other biochemical, somatometric or nutritional variables have been quantified, apart from those indicated in the study, which could be related to abdominal obesity and METs.

## 5. Conclusions

Given the beneficial association between consuming a serving of nuts three or more times per week and a lower prevalence of abdominal obesity and MetS, dietary recommendations to increase nut consumption in older people would be advisable and public health nutrition programs should implement the intake of nuts as a part of a healthy diet in the elderly population.

Further cross-sectional and longitudinal research related to nut consumption in older people should be conducted in larger communities, and in elderly populations living independently and in institutions.

## Figures and Tables

**Figure 1 ijerph-19-01256-f001:**
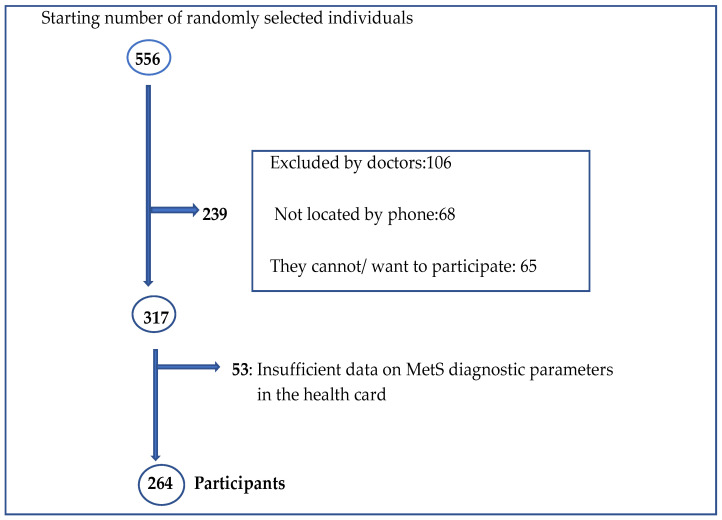
Flow chart of the participant selection process.

**Table 1 ijerph-19-01256-t001:** Nut consumption by sociodemographic and health-related variables.

	Low Nut Intake (<3 Ration/Week) (N = 158)	High Nut Intake (≥3 Ration/Week) (N = 106)	
	*n* (%)	*n* (%)	*p*-Value ^1^
**Sex**			0.388
Men	65 (63.1)	38 (36.9)	
Women	93 (57.8)	68 (42.2)	
**Age groups**			0.899
65–69	51 (58.0)	37 (42.0)	
70–74	60 (61.2)	38 (38.8)	
75–79	47 (60.3)	31 (39.7)	
**Marital status**			0.744
Married/partnered	97 (58.1)	70 (41.9)	
Separated	9 (56.3)	7 (43.8)	
Widowed	32 (66.7)	16 (33.3)	
Single	20 (60.6)	13 (39.4)	
**Type of cohabitation**			0.803
Couple	96 (58.5)	68(41.5)	
With relatives	16 (55.2)	13 (44.8)	
With a carer	2 (66.7)	1 (33.3)	
Alone	43 (64.2)	24 (35.8)	
Shared flat	1 (100)	0 (0.0)	
**Educational level**			0.399
University	58 (54.2)	49 (45.8)	
Secondary school	47 (63.5)	27 (36.5)	
Primary school	50 (65.8)	26 (34.2)	
Incomplete	3 (42.9)	4 (57.1)	
**BMI categories**			0.035
Normal weight	31 (48.4)	33 (51.6)	
Overweight	79 (59.8)	53 (40.2)	
Obesse	48 (70.6)	20 (29.4)	
**Health status**			
Hypertension	128 (61.0)	82 (39.0)	0.471
Hyperglycemia	52 (61.9)	32 (38.1)	0.642
Hipertriglyceridemia	43 (78.2)	12 (21.8)	0.002
Los HDL-c	30 (57.7)	22 (42.3)	0.723

¹ Differences between low and high nut intake were evaluated using the Pearson Chi square test.

**Table 2 ijerph-19-01256-t002:** Prevalence of metabolic syndrome by BMI and gender.

BMI	Men (N = 103)	Women (N = 161)	*p*-Value ¹	Total *p*-Value ² (N = 264)
	*n* (%)	*n* (%)		*n* (%)
Normal weight	4 (30.8)	7 (13.7)	<0.001	11 (17.2)
Overweight	25 (40.3)	27 (38.6)	0.783	52 (39.4)
Obesity	20 (71.4)	23 (57.5)	0.023	43 (63.2)

¹ Differences between gender were evaluated using the Pearson Chi square Goodness-of-Ft test; ² Chi square trend: <0.001.

**Table 3 ijerph-19-01256-t003:** Association between nut consumption and abdominal obesity and metabolic syndrome.

	Low Nut Intake (<3 Ration/Week) (N = 158)	High Nut Intake (≥3 Ration/Week) (N = 106)		
	*n* (%)	*n* (%)	PR; (95% IC)	*p*-Value ¹
Abdominal obesity	133 (84.2)	75 (70.8)	1.19 (1.03–1.37)	0.015
Metabolic syndrome	74 (46.8)	32 (30.2)	1.61 (1.16–2.25)	0.005

¹ Wald chi-square. PR: prevalence ratio. IC: 95% confidence interval.

## Data Availability

The data presented in this study are available on request from the corresponding author.

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
