# Peer review of "The Regular Consumption of Nuts Is Associated with a Lower Prevalence of Abdominal Obesity and Metabolic Syndrome in Older People from the North of Spain"

_ijerph, 2022, doi:10.3390/ijerph19031256_

Round 1
Reviewer 1 Report
Firstly, this article needs extensive editing and improvement in English language throughout the manuscript as it is difficult to understand methodologies and outcomes in places, including eg. the title: should be"....associated WITH a lower prevalence of abdominal obesity....."
Although the study yields potentially important public health messages in relation to nut consumption and health, it is difficult to ascertain the strength of these conclusions as it is difficult to comprehend how these conclusions relate to the results and if the methodology is appropriate due to confusing use of language.
The conlusion of the study finding that the consumption of a portion of nuts < 3 times per week was associated with a doubled risk of developing abdominal obesity and MetS compared to consumption ≥ 3 times a week is a strong one but were confounding factors accounted for? Was covariate analysis carried out?
Additional points
Abstract: is participant age mean and sd?
Sample size calculation is difficult to comprehend - please clarify.
IDF definiton for MEtS should be stated in methods
Thresholds for different BMI levels should be included in Methods
Results: the phrase "a significant difference (p=0.002) was only observed in the case of individuals diagnosed with hypertriglyceridemia, most of whom consumed less nuts than recommended" is relatively meaningless. What do you mean by significant difference - lower consumption?
Table 1 - which BMI threshold does the P-value represent?
Table 2 and 3- I don't see the point of these tables. How are these related to nut consumption? The focus of this article is nut consumption and prevalence of METs etc...these tables do not add to that study aim.
Section 3.5 - when you are discussing associations, you should state the strength of the association.
Author Response
Reviewer 1
All corrections are written in red:
- The title “….associated WITH…” has been modified.
- The article has been sent to MDPI translation service to improve the English edition.
- It has been added as a limitation that the association of nut consumption with abdominal obesity and METs has not been controlled for confounding variables (discussion: at the end of section 4.3)
- In the abstract, the abbreviation SD has been added to clarify that it is the standard deviation.
- The sample size justification has been modified (section 2.2)
- Two new sections have been added in material and methods:
- The definition of BMI levels with their corresponding threshods (section 2.4)
- The definition of METs according to the criteria of the International Diabetes Federation (IDF) (section 2.5)
- In the results (section 3.1) the drafting of the higher prevalence of hypertriglyceridemia in individuals consuming less than 3 servings of nuts per week compared to those consuming ≥ 3 servings per week has been modified.
- Table 1: Pearson´s chi-square test (independence of rows and columns) indicates when is significant (p=0.035) that there are an association between the variables of nut consumption and BMI categories. In addition, there is a significant trend of increasing prevalence of consumption of < 3 servings of nuts per week as BMI increases (p for trend =0.010). The results obtained with SPSS are attached.
- Tables 2 and 3 are placed at the end of the article, after references, as supplementary material.
- Section 3.5: The words: “…. Associated with…” are deleted. The strength of the association is indicated by OR.
Reviewer 2 Report
I recommend adding other methods to the article, which were determined in the monitored population. The study monitors very few biochemical, somatometric, but also nutritional parameters.
Author Response
Reviewer 2
All corrections are written in red:
- Two new sections have been added in material and methods:
- The definition of BMI levels with their corresponding threshods (section 2.4)
- The definition of METs according to the criteria of the International Diabetes Federation (IDF) (section 2.5)
- The lack of monitoring of biochemical, somatometric and nutritional parameters has been added as a limitation (in discussion: at the end of section 4.3).
Reviewer 3 Report
The manuscript is about the regular consumption of nuts is associated with a lower prevalence of abdominal obesity and metabolic syndrome in older people from the north of Spain. The ability of nuts to improve the conditions of metabolic syndrome (MetS) is now well established. The peer-reviewed work confirms the results of other researchers. The article is well structured and presents consistent arguments and theories. The cited references are current.
I suggest providing more information on the important role of fatty acid (primarily ALA (Alpha-Linolenic Acid) from nuts.
Author Response
Reviewer 3
All corrections are written in red:
-Has been provided information on the important role of omega-3 fatty acids (ALA) from nuts.
Three new bibliographic references with this information are included in the following paragraphs:
- Introduction: at the end (reference 15)
- Discussion: at the beginning section 4.1 (reference 30)
- Discussion: at the end section 4.3 (reference 72)
Round 2
Reviewer 2 Report
I appreciate the reworking of the article, which I recommend publishing in the given version. The article has become clearer and especially more valuable for the professional public.
Author Response
Dear Review 2,
Thanks a lot for your comments to improve the article.
Best regards,
Iñaki
